# Copper Uptake and Its Effects on Two Riparian Plant Species, the Native *Urtica dioica,* and the Invasive *Fallopia japonica*

**DOI:** 10.3390/plants12030481

**Published:** 2023-01-19

**Authors:** Daniel Schmitz, Johanna Girardi, Jellian Jamin, Mirco Bundschuh, Benedict Geng, Rico Feldmann, Verena Rösch, Kai Riess, Jens Schirmel

**Affiliations:** 1iES Landau, Institute for Environmental Sciences, University of Kaiserslautern-Landau (RPTU), 76829 Landau, Germany; 2Department of Aquatic Sciences and Assessment, Swedish University of Agricultural Sciences, 75007 Uppsala, Sweden; 3Eusserthal Ecosystem Research Station, University of Kaiserslautern-Landau, 76829 Landau, Germany

**Keywords:** inorganic pollution, bioaccumulation, contamination, copper, copper sulphate, heavy metals, plant growth

## Abstract

Copper accumulating in stream sediments can be transported to adjacent riparian habitats by flooding. Although being an essential element for plants, copper is toxic at high concentrations and restricts, among other things, plant growth. Besides copper, invasive plants, such as *Fallopia japonica*, which are known to be tolerant toward heavy metals, modify riparian habitats. If the tolerance of *F. japonica* is higher compared to native plants, this could accelerate invasion under high heavy metal stress. Therefore, we aimed to compare the effect of copper on two common riparian plants, the invasive *F. japonica* and the native *Urtica dioica*. We performed a pot experiment with a gradient from 0 to 2430 mg kg^−1^ of soil copper. We hypothesized that (i) negative effects on plant growth increase with increasing soil copper concentrations with *F. japonica* being less affected and (ii) accumulating higher amounts of copper in plant tissues compared to *U. dioica*. In support of our first hypothesis, growth (height, leaf number) and biomass (above- and belowground) of *F. japonica* were impacted at the 810 mg kg^−1^ treatment, while the growth of *U. dioica* was already impacted at 270 mg kg^−1^. Due to 100% mortality of plants, the 2430 mg kg^−1^ treatment was omitted from the analysis. In contrast, chlorophyll content slightly increased with increasing copper treatment for both species. While *U. dioica* accumulated more copper in total, the copper uptake by *F. japonica* increased more strongly after exposure compared to the control. In the 810 mg kg^−1^ treatment, copper concentrations in *F. japonica* were up to 2238% higher than in the control but only up to 634% higher in *U. dioica*. Our results indicate that *F. japonica* might be able to more efficiently detoxify internal copper concentrations controlling heavy metal effects compared to the native species. This could give *F. japonica* a competitive advantage particularly in polluted areas, facilitating its invasion success.

## 1. Introduction

Since the 19th century, the use of copper as a fungicide has been an inherent part of agriculture [1]. Copper does not degrade but accumulates in soils leading to elevated concentrations [2]. In fact, the long-term application of copper resulted in high levels—reaching up to 3000 mg kg^−1^—in agricultural areas [3]. In European vineyards, for example, soil copper levels of up to 600 mg kg^−1^ have been observed, while corresponding uncontaminated sites ranged between 20 and 100 mg kg^−1^ [4]. From agricultural soils, copper can be transported to adjacent non-target aquatic ecosystems through spray drift, groundwater drainage, and surface run-off [5]. In rivers and streams, copper tends to accumulate in sediments [6] where it can be remobilized and transferred to riparian ecosystems during flooding with potential effects on local plant species, affecting ecosystem structure and functioning [7]. 

Although copper is an essential element supporting enzyme activity and chlorophyll production [8], at high concentrations it is toxic, causing oxidative stress that can lead to DNA damage [9]. Moreover, copper can induce cell membrane ruptures and abnormalities in leaf chloroplasts, which impacts photosynthesis and nutrient transport [10,11]. Protective processes, which vary between species, have evolved including detoxification using enzymatic and non-enzymatic antioxidants as well as sequestration of excess copper [9]. Nonetheless, plants often respond to heavy metal exposure—such as copper—by growth impairments. For example, CuSO_4_ concentrations of 100 µM caused reduced growth rates and necrosis in *Prunus cerasifera* [12]. Copper toxicity manifesting as chlorosis and subsequent desiccation and death of plant parts as well as stunted growth has been observed in woody ornamentals [13], which is similar to symptoms produced by other heavy metals. Bouazizi et al. [14] also reported a reduction in biomass and leaf number in *Phaseolus vulgaris*.

Besides copper, plant invasion can be an additional pressure in riparian ecosystems. Common nettle (*Urtica dioica* L., Urticaceae) is a native, rhizomatous, perennial herb that often dominates in nitrogen-rich soils [15]. Its stands are frequently invaded by the Asian knotweed (*Fallopia japonica* (Houtt.) Ronse Decr., Polygonaceae), one of the most widespread invasive plants in central Europe. It is a perennial, rhizome-forming species which quickly produces high above and below ground biomass [16]. Consequently, *F. japonica* outcompetes native plants for resources such as light and nutrients by forming dense stands and altering the soil nitrogen composition, finally reducing biodiversity [17]. The success of invasive plants is often related to a higher tolerance toward heavy metals, relative to native species [18]. In support of this general statement, members of the genus *Fallopia* have been shown to be tolerant to heavy metal contamination including copper [19], which is accumulated mainly in roots (*F. convolvulus* [20], *F. japonica* [21,22,23]). While copper at concentrations up to 200 mg kg^−1^ did not affect the ability of *F. japonica* to establish from rhizomes, at concentrations equal to or above 300 mg kg^−1^ it delayed plant growth and impacted morphology [23]. This high tolerance supports the species’ competitive advantage in soils contaminated with heavy metals [24]. The co-occurring native species *U. dioica* is also considered to be tolerant to heavy metals [25], including copper [26,27], which is often accumulated in aboveground parts such as leaves [28]. However, information on the tolerance to higher soil copper concentrations is scarce. Therefore, the objective of our study was to investigate if the native plant (*U. dioica*) shows a different response to copper contamination than the invasive plant (*F. japonica*). Such a difference could impact the invasive success of *F. japonica* which is a major factor in riparian ecosystems.

To test if there is such a difference in response, we conducted a pot experiment under field conditions. In a randomized block design, the plants were grown in soils with increasing soil copper concentrations (0 to 2430 mg kg^−1^ soil). We expected that (i) higher copper concentrations reduce plant growth of both species. However, since *F. japonica* is considered to be more tolerant, this invasive species should only respond at higher copper concentrations compared to *U. dioica*. We further hypothesized, based on the known ability of *F. japonica* to accumulate heavy metals, that (ii) the invasive species exhibits higher copper concentrations and takes up copper more effectively in all plant parts compared to *U. dioica*.

## 2. Results

Since no plants survived in the highest copper treatment (2430 mg copper kg^−1^ soil) (see Appendix A), this treatment was disregarded for further analysis. In the text, all mentioned values will be mean and standard deviation (see Section 4.4 for more details).

### 2.1. Seasonal Plant Growth

Plant height and leaf number significantly decreased with increasing copper concentrations in both *Fallopia japonica* and *Urtica dioica*. In the control treatment, *F. japonica* (26.8 ± 12.9 cm) was about 25% higher than in the treatment with 810 mg kg^−1^ (21.4 ± 12.6 cm), which was statistically significant (Table 1). Similarly, in *U. dioica* the average height was significantly higher in the control (35.7 ± 22.9 cm) than at 810 mg copper kg^−1^ (30.0 ± 22.3), a reduction by about 20%. Regarding leaf number, *F. japonica* had an average of 56.2 ± 37.4 leaves in the control, a value significantly lower by 25% (43.0 ± 26.0) in the 810 mg copper kg^−1^ treatment. For *U. dioica*, a significant reduction in leaf number from 140.0 ± 131.9 to 73.6 ± 72.2 (by approximately 50%) in the same treatments was also observed (Table 1). Overall, the decrease in plant height and leaf number became apparent at 810 mg copper kg^−1^ for *F. japonica* while *U. dioica* showed significant responses already at 270 mg copper kg^−1^ (Figure 1).

In contrast, the chlorophyll content increased statistically significantly with increasing copper concentration (Table 1). For *F. japonica*, chlorophyll increased by about 10% from 21.1 ± 5.7 in the control to 23.1 ± 6.0 in the 810 mg copper kg^−1^ treatment. With an increase in chlorophyll from 27.3 ± 7.2 to 30.6 ± 13.1, the effect was comparable for *U. dioica*.

Apart from treatment effects, the factor time also significantly influenced all plant parameters. Height and leaf number of both species increased during spring but remained rather constant thereafter (Figure 1). In contrast, the chlorophyll content significantly decreased over time independent of the plant species (Table 1). Both copper treatment and time significantly interacted for several variables in both species (Table 1). At lower copper concentrations, leaf number of *F. japonica* increased more strongly over time while higher treatment levels showed a decrease in the later part of the season. Height, a parameter describing growth over time, of *U. dioica* was lower at higher copper concentrations while leaf numbers decreased more strongly later in the growing season (Figure 1).

### 2.2. Biomass after a Two-Year Experiment

The final biomass of *F. japonica* for both above- and belowground plant parts significantly decreased with higher copper concentrations, which became statistically significant at 810 mg copper kg^−1^ (Table 2). Above- and belowground biomass significantly decreased by a factor of two from 72.1 ± 24 g to 32.5 ± 28 g and 228.2 ± 98.5 g to 114.6 ± 126 g, respectively. At lower copper concentrations, biomass remained largely constant relative to the control (Figure 2a,c).

For *U. dioica*, there was a tendency for a decrease in above- and belowground biomass with increasing copper concentrations but no significant effect overall (Table 2). Above- and belowground biomass decreased by a factor of two and three respectively from 29.9 ± 19.1 g to 12.6 ± 11.3 g and 54.3 ± 30.1 g to 17 ± 18.8 g. The statistical non-significance of the copper treatments in *U. dioica* may be explained by a rather low biomass in the control, with the lowest copper treatment of 90 mg kg^−1^ having higher biomass than the control (Figure 2b,d).

### 2.3. Copper Content of Plant Tissues

The copper content of almost all plant tissues was higher in *U. dioica* compared to *F. japonica* across the experimental soil copper gradient. The only exception was the root copper content, which was 46% higher in *F. japonica* at the 810 mg kg^−1^ level (17.1 ± 9.5 mg g^−1^ in *F. japonica* and 11.7 ± 2.1 mg g^−1^ in *U. dioica*). Especially for the aboveground and rhizome material, the samples from *U. dioica* showed much higher copper contents compared to *F. japonica*. In the 810 mg kg^−1^ treatment, *U. dioica* aboveground material contained about 2.5 times more copper compared to *F. japonica* (9.7 ± 5.9 mg g^−1^ in *F. japonica* and 24 ± 11.9 mg g^−1^ in *U. dioica*) but aboveground copper content was higher in *U. dioica* across all treatment levels (Figure 3a,b).

Copper content of the three sample types (aboveground, roots, and rhizome) significantly increased with increasing soil copper concentration in both *F. japonica* and *U. dioica* (Table 3). Like the absolute concentrations, the strongest increase was found in the rhizome where the mean copper content increased from 1.3 ± 1.0 mg g^−1^ in the control to 30.4 ± 31.9 mg g^−1^ in the 810 mg kg^−1^ level for *F. japonica* and from 9.4 ± 13.5 mg g^−1^ to 69.0 ± 33.9 mg g^−1^ in *U. dioica* (Figure 3e,f). Unlike the absolute concentrations, *F. japonica* showed a much higher accumulation compared to the control, increasing copper concentrations in rhizomes by more than 23.3 times compared to 7.3 times in *U. dioica*. Accumulation in the 810 mg kg^−1^ treatment compared to the control for the two other plant parts was also higher in *F. japonica* compared to *U. dioica*. For aboveground samples, copper content increased by 19 times in *F. japonica* vs. 17 times in *U. dioica* and for the roots, the difference was 3.5 times compared to 2 times.

In *F. japonica*, the highest copper content in tissues was found in the rhizome, with a two-fold higher copper concentration at the highest treatment level compared to root samples (30.4 ± 31.9 mg g^−1^ compared to 17.1 ± 9.5 mg g^−1^) and a three-fold higher concentration compared to aboveground samples (30.4 ± 31.9 mg g^−1^ compared to 9.7 ± 5.9 mg g^−1^). The rhizome copper concentration increased by 2238% from the control (1.3 ± 1 mg g^−1^) to the 810 mg kg^−1^ treatment. For *U. dioica*, the average copper content was also highest in rhizomes (six times higher than roots and three times higher than aboveground material at the highest treatment; Figure 3). For *U. dioica*, the increase from control (9.4 ± 13.5 mg g^−1^) to the highest treatment was 634%.

## 3. Discussion

We investigated the effect of copper contamination (treatments with 0, 90, 270, and 810 mg kg^−1^) on the invasive plant *Fallopia japonica* and the native *Urtica dioica*. Higher soil copper concentrations negatively impacted the growth of both the invasive plant species *Fallopia japonica* and the native *Urtica dioica*. However, the native species was more sensitive relative to the invasive species. Uptake of copper by plant tissue increased with soil copper content for both plants. Unexpectedly, *U. dioica* showed higher absolute content of copper in plant material compared to *F. japonica*. However, the accumulation of copper compared to the control was higher in *F. japonica* than in *U. dioica*, which corresponds to the hypothesis that *F. japonica* can accumulate copper more effectively.

### 3.1. Copper Contamination Negatively Impacts Plant Growth, but More Strongly in U. dioica Than F. japonica

As expected, increasing copper concentrations impacted all observed plant parameters (above- and belowground biomass, plant height, leaf number and chlorophyll content) in both plant species. We also found that the growth of *F. japonica* and *U. dioica* depended on the time of measurement. Plant height and leaf number increased during the growing season due to seasonal progression in plant growth. Regarding the copper treatment, we found that biomass, plant height, and leaf number decreased with increasing copper concentration. This reduction in plant growth is in line with the previous studies on various species (*Buxus sempervirens, Cotoneaster divaricatus* and *Rhododendron obtusum* [13]; *Phaseolus vulgaris* [14]; *Prunus cerasifera* [12]) and reflects the general known responses of plants to copper [9]. The impact of copper on plant growth can be explained by oxidative damages caused by excess copper in cells, which damage cell organelles and DNA [9,10,11]. This leads to reduced growth and necrosis in affected plants [12,13,14].

Importantly, we found that the reduction in growth occurred at different threshold concentrations: For *F. japonica*, growth was only lower at the highest analyzed concentration (810 mg kg^−1^), while *U. dioica* already showed significant loss of growth at 270 mg kg^−1^. Our results therefore only partly confirm the few existing studies on *Fallopia* species, where negative effects of copper started at lower concentrations than in our study. For example, Sołtysiak [23] found impact on *F. japonica* at 300 mg copper kg^−1^ and Pedersen et al. [20] estimated a threshold for the onset of adverse effects on *Fallopia convolvulus* growth at slightly lower concentrations (i.e., 200 mg kg^−1^). In our study, plants grown at 270 mg copper kg^−1^ had similar biomass compared to the lower treatments. Although we observed the highest above- and belowground biomass in *U. dioica* at 90 mg copper kg^−1^ and a subsequent decrease toward the higher concentrations, the overall effect of copper was not significant, likely due to the high variability in our data. The stimulation of above- and belowground biomass in *U. dioica* corresponds well to the higher leaf numbers and height in the same treatment. In earlier studies, *U. dioica* has shown the ability to tolerate higher concentrations of copper with phytostabilisation of contaminated soil, meaning a reduction in the bioavailability of copper in the soil [26]. The pattern observed in our study might be due to hormesis, where a mild stress, here induced by copper, can stimulate plant growth, which might be linked to the production of antioxidants in combination with a stimulation of chlorophyll levels [29]. This effect is often observed in hyperaccumulator plants [30]. A possible reason for the stronger growth at high copper concentrations in *F. japonica* might be its higher tolerance to copper [19], which is thought to be connected to the plant’s ability to sequester copper into the cell wall [31] and produce specific binding proteins even at high metal concentrations [32]. This also allows *F. japonica* to hyperaccumulate metals [33,34]. Our findings therefore support the hypothesis that invasive plants benefit from disturbances by heavy metal contamination due to their higher tolerance to stressors such as copper [18].

In contrast, the chlorophyll content increased with copper concentration in both species and decreased along the time of measurement, since chlorophyll production is reduced in order to focus on production of flowers and seeds later in the year. According to Kalaji et al. [35], the effect of heavy metal contamination on the number of photosynthetic pigments is not uniform. For example, lead caused a decrease in the amount of pigment in *Pisum sativum* [36]. Similarly, chromium contamination decreased total chlorophyll by up to 61% in *Brassica juncea* [37] and by 54% in *Solanum lycopersicum* [38]. On the other hand, cadmium had no impact on the amount of photosynthetic pigments in *Brassica napus* but still disrupted photosynthetic processes [35]. However, in mosses (*Ptychanthus striatus*, *Thuidium delicatulum,* and *T. sparsifolium*) as well as in *Empetrum nigrum* and *Camellia sinensis*, copper showed a stronger inhibitory effect on total chlorophyll than other heavy metals, due to the disruption of chlorophyll production [39,40,41]. The contrary pattern observed in our study could be explained by substantial differences in plant traits. Both *F. japonica* and *U. dioica* are known to accumulate heavy metals [20,42] and might therefore be efficient in detoxifying copper before it can damage chlorophyll production. This might be due to chelating agents such as oxalic acid which can bind heavy metals and reduce toxicity. This has been shown for chromium toxicity [37,43]. Another factor might be a higher concentration of the plant hormone salicylic acid in these species, which can increase the stress tolerance of plants. Application of salicylic acid can decrease the toxicity of heavy metals such as chromium and aluminum [38,44]. A further explanation might be that the observed loss of leaves in higher copper treatments caused an increase in chlorophyll production in the remaining leaves to keep the photosynthetic activity of the organism stable. This form of compensation has previously been found as a response to herbivory [45,46].

### 3.2. Copper Content in Plant Tissue Is Higher in U. dioica Than in F. japonica

It could be verified that both *F. japonica* and *U. dioica* accumulate copper, which is in line with the studies dealing with roots and rhizomes [21,28,31,42]. We also found that the copper content was highest in the belowground plant parts, especially rhizomes. This observation contrasts with literature highlighting higher heavy metal concentrations in aboveground plant parts (mainly leaves) for *F. japonica* [23] and *U. dioica* [26,28]. Moreover, our study points to a lower copper concentration in the plant relative to the surrounding environment (i.e., soil) for both species, suggesting they are not hyperaccumulators of copper, which was postulated for *F. japonica* [47] and *Fallopia × bohemica* [48]. However, copper hyper accumulation was reported for roots of *F. convolvulus* with a factor 2:1 for root: soil [20].

Unexpectedly, the absolute copper concentrations in *F. japonica* were lower in nearly all copper treatments and plant parts than in *U. dioica*. This difference might be explained by *F. japonica* being able to regulate the copper uptake compared to *U. dioica*. Lerch et al. [49] showed in a pot experiment that the transfer of copper from soil to plant does not increase strongly under copper contamination. While copper in roots of *F. convolvulus* increased linearly with increasing soil copper concentration, a saturation in aboveground plant parts was reported for 200 to 300 mg kg^−1^ [20]. Only in our treatment with a copper concentration of 810 mg kg^−1^, less copper was detectable in roots of *U. dioica* compared to *F. japonica* (Figure 3). Since saturation in copper uptake was observed for *F. convolvulus* only for aboveground plant parts [20], this could also provide an explanation for the observation of increased copper content in roots of *F. japonica*. Furthermore Lu et al. [50] demonstrated a higher concentration of Fe, Al, and Cu in roots compared to rhizomes, stems, and leaves in *F. sachalinensis*. This accumulation in roots of *Fallopia* spp. is consistent with both our hypothesis and literature where *F. japonica* was found to have a high potential to accumulate heavy metals under field conditions and elevated heavy metal concentrations [21,22,23,50]. Metal accumulation is linked to metal tolerance and is common in plants found in metalliferous habitats [51]. The higher metal tolerance in *F. japonica* found in this study might therefore be linked to this ability.

## 4. Materials and Methods

### 4.1. Experimental Setup

The mesocosm experiment was set up on an open field located in Siebeldingen at the Julius Kühn-Institut (Federal Research Centre for Cultivated Plants), Rhineland-Palatinate, Germany (49.219643, 8.049299). Pots were established in April 2020 and the experiment lasted until September 2021 for two full growing seasons. In 2020, the average temperature was 11.9 °C and the total precipitation was 630 mm, while in 2021, the average temperature was 10.4 °C and the total precipitation was 814 mm [52].

In April 2020, 100 pots with a diameter of 63 cm were filled with 5 cm of gravel at the bottom as a root drainage and a mixture of 70 liters of sand originating from locally sourced Triassic sandstone (BVG, Albersweiler, Germany) and 20 liters of natural riparian soil. The riparian soil (topsoil 0–25 cm) including its natural microbial community was collected in March 2020 from a nearby unpolluted riparian area within the biosphere reserve Palatinate Forest (Dürrentalbach, 49.255048, 7.939021). Before mixing, the soil was sieved and visible plant parts were removed. While filling the pots with soil, 10 g of nitrogen fertilizer, N-P-K: 6-6-9 (ORGASAN, Cuxin DCM, Telgte, Germany) and 9 g of trace element fertilizer (Micromax Premium, Everris International, de Heerlen, The Netherlands) were added to each pot. The fertilization was repeated in the second growing season in April 2021. The final average soil pH was 7.5.

The pots were arranged in a two-factorial randomized block design (Figure 4). Plant species (*F. japonica* and *U. dioica*) and soil copper concentration were used as treatment factors. Five levels of CuSO_4_ (Centrum Metal Odczynniki Chemiczne, Falenty, Poland) in 5 liters of tap water solution were applied and mixed with the soil. This compound was chosen since it is a common fungicide utilized in agriculture. The four nominal concentrations were 0, 90, 270, 810, and 2430 mg Cu kg^−1^ soil. Each treatment combination was replicated ten times.

Nominal copper concentrations were verified using inductively coupled plasma—optical emission spectrometry (ICP-OES) (Agilent 720 ICP OES, Agilent Technologies, Santa Clara, CA, USA), following digestions with aqua regia [53,54] directly after test initiation (April 2020) and at its termination (August 2021). Measurements suggest adequate spiking with deviations relative to nominal concentrations not exceeding 15%, justifying the use of the latter throughout the present document (see Table 4). The full results of the ICP-OES measurements can be found in Appendix A.

Plants were introduced into the pots in June 2020 as rhizomes with one aboveground shoot. Rhizomes of both species originated from a riparian zone at the river Queich in Landau (49.198357, 8.096627). This stand of *F. japonica* was selected since it is genetically pure without the hybrid influence of *F. sachalinensis* (*F. Schmidt* ex Maxim.) [55]. Rhizomes were washed and cut to approximately 5 cm, making sure that at least one node was included for regeneration. Four rhizomes of the invasive or the native species, respectively, were evenly distributed in each pot.

During the growing season, the plants were watered depending on the weather conditions. Under hot and dry conditions, each pot received four liters of tap water at a maximum rate of twice a week. After precipitation events no water was provided. Randomly growing other plant species were weeded every week to minimize their impact. During the first season, the test system equilibrated and plant-related measurements started in spring of the second season. Pots without plant growth were excluded from analyses (see Appendix A for an overview of plant mortality in the experiment).

### 4.2. Plant Parameter Measurements

Plant parameters were measured bi-weekly covering ten timepoints between April and August 2021 (Appendix A). These parameters included plant height, leaf number, and chlorophyll content. Plant height (cm) was defined as size from the soil surface to the extended tip of the highest stem using a measuring stick. The number of leaves were counted. Chlorophyll content, a proxy for photosynthetic activity [56], was measured using a SPAD-502 device (Konica Minolta, Tokyo, Japan) on three randomly chosen healthy leaves per plant. Measurements with the SPAD-502 produce relative values that are proportional to the amount of chlorophyll present in the leaf [57]. The average of these measurements for each pot entered further statistical analyses. In September 2021, plants were harvested individually, separating above- and belowground biomass (Appendix A). All plant material was dried for 4 days at 60 °C. The dry biomass was then quantified using a KERN ADJ 200-4 scale (linearity ± 0.4 mg, Kern & Sohn, Balingen, Germany). In the subsequent statistical analyses, total biomass per pot separated into above- and belowground biomass was used.

### 4.3. Plant Copper Content

Dried plant material of aboveground parts (stems and leaves), roots, and rhizomes was ground into a powder separately. For each plant species and the three plant parts selected, five samples of each copper treatment were prepared, resulting in a total of 120 samples. Plant copper content was measured using ICP-OES (Agilent 720 ICP OES, Agilent Technologies, Santa Clara, CA, USA). Measurement of copper was done at a wavelength of 327.395 nm using the same method as the soil copper measurement [53,54]. Prior to analyses, samples were digested in 10 mL aqua regia overnight and afterwards heated to 175 °C for 15 min in a microwave (Mars 5 System, CEM Corporation, Matthews, North Carolina, USA). The samples remained in the microwave at 175 °C for another hour. Aliquots were subsequently diluted to 1:10 or 1:20 with MiliQ water depending on the limits of quantification supporting a reliable quantification of copper. The measured copper content in the plant parts can be found in Appendix A.

### 4.4. Data Analysis

R version 4.0.5 [58] was used for statistical analyses. Linear mixed-effects models (lmer) were created separately for the two plant species (*F. japonica* and *U. dioica*) using the package “lme4” [59]. The highest copper treatment (2430 mg kg^−1^ soil) was disregarded for further analysis because no plants survived (Appendix A). The models for each dependent variable (above- and belowground final biomass, plant height, leaf number, chlorophyll content, plant copper content) used the copper levels (factor with the four levels “0”, “90”, “270”, “810” mg kg^−1^) as a fixed effect. In the models for plant height, leaf number, and chlorophyll content, time (factor with 10 time points over the growing season) and the interaction of time and copper treatment were included as additional fixed effects. In all models, the random effect “block” was used to account for the block design and possible variability introduced by the position on the experimental site. Additionally, the random effect “stem number”, representing the total number of plant individuals found in each pot, was used to account for differences in growth caused by the uneven number of plant individuals that survived until the start of measurement in 2021. Significance levels were calculated using the ANOVA command from the “car” package [60].

## 5. Conclusions

Overall, we found that soil copper contamination inhibited the growth in terms of leaf number, height, and biomass in both *Fallopia japonica* and *Urtica dioica* pointing to a higher sensitivity of *U. dioica* (significant effects starting with 270 mg copper kg^−1^) compared to *F. japonica* (significant effects starting at 810 mg copper kg^−1^). On the other hand, *F. japonica* contained far lower absolute concentrations of copper in its above- and belowground parts compared to *U. dioica* but was able to accumulate copper more effectively when exposed to it. In this way, *F. japonica* may gain a selective advantage against this commonly occurring native species in its invasive range and especially in polluted areas because of its higher tolerance to copper. Our study demonstrated that disturbances by agricultural activity, one major source of copper pollution, could have an impact across much wider distances than expected by facilitating plant invasion success of *F. japonica* in previously undisturbed areas.

## Figures and Tables

**Figure 1 plants-12-00481-f001:**
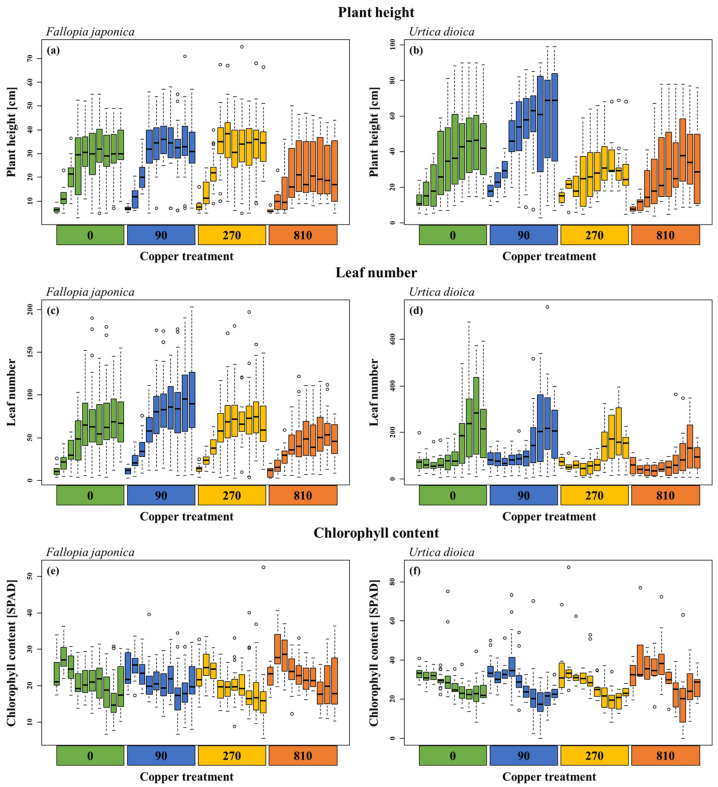
Plant growth parameters of *Fallopia japonica* and *Urtica dioica* recorded bi-weekly across the growing season (n = 10 time points). Shown are boxplots with median values represented by the black line. Each color represents one copper treatment from control to 810 mg kg^−1^. Within each treatment, the time in the season is shown from left to right. (**a**) Plant height of *Fallopia japonica*, (**b**) plant height of *Urtica dioica*, (**c**) leaf number of *Fallopia japonica*, (**d**) leaf number of *Urtica dioica*, (**e**) chlorophyll content of *Fallopia japonica*, (**f**) chlorophyll content of *Urtica dioica*.

**Figure 2 plants-12-00481-f002:**
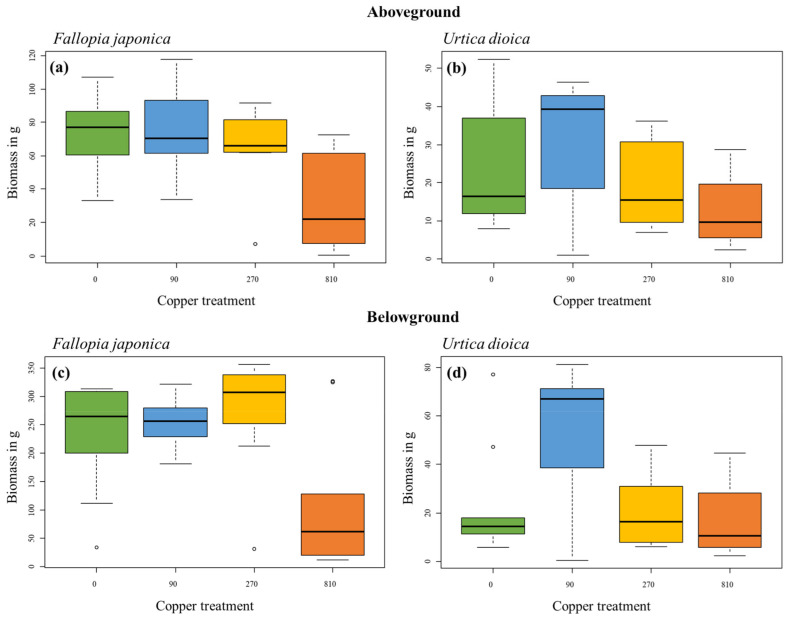
Biomass of above- and belowground plant material of *Fallopia japonica* and *Urtica dioica* collected at the end of the growing season in each copper treatment (n = 10 replicates). Shown are boxplots with median values represented by the black line. Each color corresponds to one copper treatment from control to 810 mg kg^−1^ copper. (**a**) Aboveground biomass of *Fallopia japonica*, (**b**) aboveground biomass of *Urtica dioica*, (**c**) belowground biomass of *Fallopia japonica*, (**d**) belowground biomass of *Urtica dioica*.

**Figure 3 plants-12-00481-f003:**
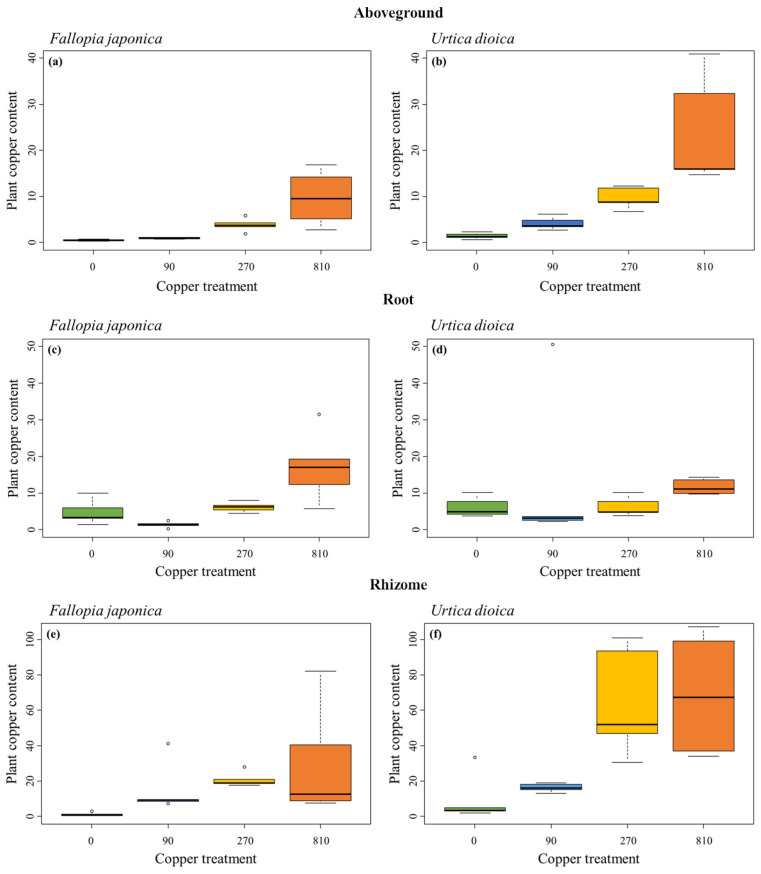
Plant copper content in mg g^−1^ for each copper treatment (0 mg kg^−1^ in green, 90 mg kg^−1^ in blue, 270 mg kg^−1^ in yellow and 810 mg kg^−1^ in orange, n = 5) in *Fallopia japonica* and *Urtica dioica*. The y-axis is scaled to the same values for aboveground, root, and rhizome samples, respectively, but differs between the three sample types. Shown are boxplots with median values represented by the black line. (**a**) Aboveground copper content in *Fallopia japonica*, (**b**) aboveground copper content in *Urtica dioica*, (**c**) root copper content in *Fallopia japonica*, (**d**) root copper content in *Urtica dioica*, (**e**) rhizome copper content in *Fallopia japonica*, (**f**) rhizome copper content in *Urtica dioica*.

**Figure 4 plants-12-00481-f004:**
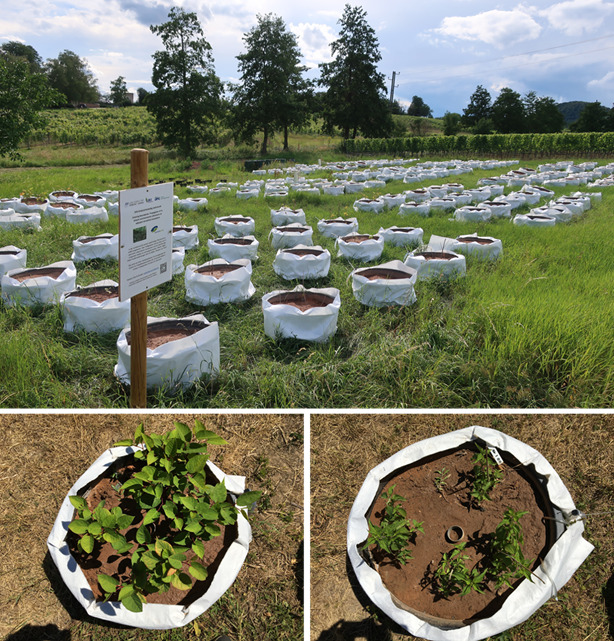
Experimental setup of the experiment. General view of the pots arranged in a randomized block design and sample pictures of *Fallopia japonica* (left) and *Urtica dioica* (right) growing in the pots in spring 2021.

**Table 1 plants-12-00481-t001:** Results of the lmer models for both *Fallopia japonica* and *Urtica dioica* performance parameters across the growing season of 2021. Chi-squared and corresponding *p*-values for all analyzed parameters are given. Treatment represents the effect of the copper treatment; time represents the effect of the time over the season when the parameter was recorded; and treatment * time represents the interaction between both parameters. Significant values below the level of 0.05 are printed in bold.

	Treatment		Time		Treatment * Time	
	Chi^2^	*p*	Chi^2^	*p*	Chi^2^	*p*
*Fallopia japonica*						
Plant height	77.37	**<0.001**	209.69	**<0.001**	3.89	0.273
Leaf number	48.82	**<0.001**	286.16	**<0.001**	21.35	**<0.001**
Chlorophyll content	40.49	**<0.001**	220.73	**<0.001**	8.65	**0.034**
*Urtica dioica*						
Plant height	147.29	**<0.001**	383.30	**<0.001**	21.14	**<0.001**
Leaf number	35.88	**<0.001**	354.69	**<0.001**	30.85	**<0.001**
Chlorophyll content	14.73	**0.002**	232.31	**<0.001**	7.01	0.072

**Table 2 plants-12-00481-t002:** Results of the lmer models for both *Fallopia japonica* and *Urtica dioica* above- and belowground plant biomass collected at the end of the second growing season. Chi-squared and corresponding *p*-value for each analyzed parameter are given. Treatment represents the effect of the copper treatment. Significant values below the level of 0.05 are printed in bold.

	Treatment	
	Chi^2^	*p*
*Fallopia japonica*		
Aboveground biomass	34.92	**<0.001**
Belowground biomass	19.00	**<0.001**

*Urtica dioica*		
Aboveground biomass	4.17	0.244
Belowground biomass	6.59	0.086

**Table 3 plants-12-00481-t003:** Results of the lmer models for both *Fallopia japonica* and *Urtica dioica* plant copper content separated by aboveground, root and rhizome samples collected at the end of the second growing season. Chi-squared and corresponding *p*-value for each analyzed parameter are given. Treatment represents the effect of the copper treatment. Significant values below the level of 0.05 are printed in bold.

	Treatment	
	Chi^2^	*p*
*Fallopia japonica*		
Aboveground samples	168.89	**<0.001**
Root samples	47.16	**<0.001**
Rhizome samples	44.57	**<0.001**

*Urtica dioica*		
Aboveground samples	143.79	**<0.001**
Root samples	41.56	**<0.001**
Rhizome samples	65.84	**<0.001**

**Table 4 plants-12-00481-t004:** Total copper found in the soil in mg kg^−1^ in relation to the desired treatment. See Appendix A for the corresponding data.

Treatment [mg kg^−1^] (April 2020)	Initial Soil Copper Content [mg kg^−1^] (April 2020)	Deviation from Treatment	Final Soil Copper Content [mg kg^−1^] (August 2021)	Deviation from Treatment
0	8.18 ± 1.35	-	4.99 ± 1.32	-
90	87.63 ± 11.62	−2.6%	93.63 ± 9.58	4.0%
270	238.99 ± 38.50	−11.5%	252.83 ± 35.89	−6.4%
810	781.64 ± 154.91	−3.5%	714.48 ± 85.42	−11.8%

## Data Availability

The authors declare that the data supporting the findings of this study are available within the article and its Appendix A.

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
