# Peer review of "Copper Uptake and Its Effects on Two Riparian Plant Species, the Native Urtica dioica, and the Invasive Fallopia japonica"

_plants, 2023, doi:10.3390/plants12030481_

Round 1

Reviewer 1 Report

Comments to the Author: Authors has taken the invasive Fallopia japonica and the native Urtica dioica that represents the findings of the copper uptake in the better way. The research findings could be published in this journal. The article presents previously unpublished data and the overall research, and the findings given by the authors are good. The research findings could be published in this journal after some minor corrections and revisions.

Some comments and recommendations are listed below:

Please improve the quality of English grammar in the text, as well as check punctuations, spaces, incomplete sentences, spellings, etc.

On what basis the higher concentration i.e., 2,430 mg kg-1 soil was selected? Explain in brief?

In results “see chapter 4.4 for more details”, use word section in place of underlined word.

Authors should check grammar very carefully throughout the MS as there are many typo and grammar mistakes.

References should be checked for spacing, italics, bold, abbreviations for journal names, and proper formatting as per journal guidelines.

Review comments:

The manuscript can be accepted, after minor revision

Comments for Editors:

The manuscript can be accepted, after minor revision

Reviewer 2 Report

REVIEW REPORT

Ms title- Copper Uptake and its Effects on Native and Invasive Riparian Plant Species

Authored by- Daniel Schmitz 1, * Johanna Girardi 1, Jellian Jamin 1, Mirco Bundschuh 1, 2, Benedict Geng 1, Rico Feldmann 1, Verena Rösch 1, Kai Riess 1 and Jens Schirmel 1, 3

1 iES Landau, Institute for Environmental Sciences, University of Koblenz-Landau, Germany 2 Department of Aquatic Sciences and Assessment, Swedish University of Agricultural Sciences, Sweden

3 Eusserthal Ecosystem Research Station, University of Koblenz-Landau, Germany

* Correspondence: schmitzda@uni-landau.de.

Submitted in: Plants

Reviewer’s recommendation: Minor Revision

Overall comments to authors:

             After the thorough analysis of the Ms, it has been projected that the main question addressed by the researcher seems to be indicating that increasing soil copper concentrations negatively impacted the growth of both the invasive plant species Fallopia japonica and the native Urtica dioica. However, the native species was more sensitive compared to the invasive species. Uptake of copper into plant tissue increased with soil copper content for both plants. Unexpectedly, U. dioica showed higher absolute content of copper in plant material compared to F. japonica. However, the accumulation of copper compared to the control was higher in F. japonica than in U. dioica, which corresponds to the hypothesis that F. japonica can accumulate copper more effectively. But the Ms is seriously missing any important parameter for comparing the accumulative capacity of the two species. There is no novelty statement which could highlight the uniqueness of the work done and also the abstract is not very potent in highlighting the importance of the Ms. The keywords are also not suitable in helping to find the article in the current registers or indexes. The topic is of general interest or not is also not clear from the Ms as the importance of Riparian plants is not elaborated and all the relevant aspects of the topic are not fully presented. Abstract is also not appropriately covering the contents of the article.

The general comments to authors are given below:

General comments-

1.      Grammatical errors are present, please revise the whole manuscript to remove any possible grammatical errors, redundancy and typos for example- a sentence in the abstract- ‘‘Although being an essential element for plants, copper is toxic at high concentrations reducing amongst others plant growth’’ is not correct grammatically. The topic of present study is fine however, the paper somewhat failed to answer some objectives of the study.

2.      Error in sentence formation, please revise the whole manuscript to avoid the use of long sentences and the paragraphs are very short.

3.      Please maintain the uniformity while citations and book referencing must be thoroughly checked.

4.      You should give a cue of your treatments in the starting of the discussion section and results must be discussed in the abstract to make it easier for the readers

5.         In the keywords, it is strongly advisable use suitable words that can aid in finding out the manuscript in current registers or indexes.

6.      Ruyters S., Salaets P., Oorts K., Smolders E. Copper toxicity in soils under established vineyards in Europe: A survey. Science of The Total Environment 2013 443: 470–477. https://doi.org/10.1016/j.scitotenv.2012.11.001, the use of ‘:’ is creating non-uniformity in the referencing pattern, please check and correct it.

7.      ‘‘Fox, J. & Weisberg, S. An {R} Companion to Applied Regression, Third Edition. Sage: Thousand Oaks, CA, 2019. URL: https://socialsciences.mcmaster.ca/jfox/Books/Companion/’’. Is this a book reference??, please cite accordingly.

Abstract and Introduction:

1.      Add results from all the sections precisely in the abstract.

2.      Keywords should be more and manuscript specific as well as unique.

3.      ‘‘We hypothesized that (i) effects on plant growth increases with increasing soil copper concentrations with F. japonica being less sensitive’’, please specify the effects are positive or negative to make it clear for the readers.

4.      Objectives of present study should be added briefly at the end of introduction. The need of this study should be incorporated in justified manner in this Ms.

Materials and Methods:

1.      ‘‘Pots were established in April 2020 and the experiment lasted until September 2021 for two full growing seasons. In 2020, the average temperature was 11.9 °C and the total precipitation was 630 mm, while in 2021, the average temperature was 10.4 °C and the total precipitation was 814’’. Please specify if the changes in temperature and total precipitation resulted in any difference in the growth of the two Riparian species.

2.      Plant copper content measurement, the protocol given in the Ms. is not clearly stating whether you use AAS for copper determination or some other method. Rewrite it and cite accurate papers suggested in the last portion of this review report.

Results:

1.      ‘‘In contrast, the chlorophyll content statistically significantly increased with increasing copper concentration (Table 1). For F. japonica, chlorophyll increased by about 10% from 21.1 ± 5.7 % in the control to 23.1 ± 6.0 % in the 810 mg copper kg-1 treatment. With an increase in chlorophyll from 27.3 ± 7.2 % to 30.6 ± 13.1 %, the effect size was comparable for U. dioica’’. If the chlorophyll is increasing as per your result, then why the growth is decreasing and why photosynthesis does not have any significant impact on the growth of plants on increasing copper, there is no clear discussion given regarding this result which is creating doubt, please specify and validate the results with previous studies which are suggested in the last portion of this review report.

2.      In the result - Copper content of plant tissues, it has been mentioned that ‘‘The only exception was the root copper content, which was 46 % higher in F. japonica at the 810 mg kg-1 level (17.1 ± 9.5 mg g-1 in F. japonica and 11.7 ± 2.1 mg g-1 in U. dioica). Especially for the aboveground and rhizome material, the samples from U. dioica showed much higher copper contents compared to F. japonica. In the 810 mg kg-1 treatment, U. dioica aboveground material contained about 2.5 times more copper compared to F. japonica’’. No clear reason is given in the Ms. for this anamoly and why U. dioica contains more copper content in the above ground parts even when the copper was supplied to the soil. Please revise it accordingly.

Discussion:

1.       ‘‘Increasing soil copper concentrations negatively impacted the growth of both the invasive plant species Fallopia japonica and the native Urtica dioica. However, the native species was more sensitive relative to the invasive species’’. Please specify why increase in photosynthesis due to increase in copper content has no impact on the plant growth.

2.      Discussion should be written in form of conclusion of the result and help in justifying the results but the discussion of the Ms. is vaguely written and more case studies can be added in the discussion. The validation of hypothesis, facts, and results essentially require with the more citation of papers which are suggested in this review report.

Conclusion:

Conclusion should be justified sufficiently as it does not contain any results from the manuscript. Please revise it accordingly.

References:

The introduction, results and discussion sections are seriously missing the validation of the results and many scientific facts and hypothesis in present form of this Ms. This should be clearly addressed and validated properly with previous studies. Therefore, the below references must be cited in introduction, material and methods, results and discussion sections of this Ms. which will improve the Ms quality significantly.

  • 1. Singh, D., Agnihotri, A., and Seth, C. S. 2017. Interactive effects of EDTA and oxalic acid on chromium uptake, translocation and photosynthetic attributes in Indian mustard (Brassica juncea L. var. Varuna). Current Science, 2034-2042.
    2. Gupta, S., and Seth, C. S. 2021. Salicylic acid alleviates chromium (VI) toxicity by restricting its uptake, improving photosynthesis and augmenting antioxidant defense in Solanum lycopersicum L. Physiology and Molecular Biology of Plants, 27(11), 2651-2664..

Reviewer 3 Report

The following should be considered when revising.

1. The title is too general.

2. The authors selected a boxplot to present their results. To my knowledge, this approch might not be inappropriate. More importantly, there were five replicates, how did they make boxplotx?

3. In the Figure 1, there were 10 data points or 11 data points. So why did the authors stated 10 time points?

4. As mentioned by the authors, leaf chlorophyll was measured with a SPAD meter. So the SPAD values had no unit. I cann't understand why the authors used percent.

5. The authors used the terms shoots and leaves. In fact, shoots include leaves. If this is true, then these terms might be incorrect.

6. In the Discussion, the authors stated that copper contamination genatively impacts plant growth. This expression is not correct because a low concentration did not influence plant growth.

Round 2

Reviewer 3 Report

I am satisfied with the revisions by the authors.